# Detection of Organosulfur and Organophosphorus Compounds Using a Hexafluorobutyl Acrylate-Coated Tapered Optical Fibers

**DOI:** 10.3390/polym14030612

**Published:** 2022-02-04

**Authors:** Karol A. Stasiewicz, Iwona Jakubowska, Michał Dudek

**Affiliations:** Institute of Applied Physics, Military University of Technology, 00-908 Warsaw, Poland; iwona.jakubowska@wat.edu.pl (I.J.); michal.dudek@wat.edu.pl (M.D.)

**Keywords:** polymers, hexafluorobutyl acrylate, organosulfur and organophosphorus compounds, optical fiber technology, tapered optical fibers, sensors

## Abstract

This paper presents the results of a study on the possibility of detecting organosulfur and organophosphorus compounds by means of polymer-assisted optical fiber technology. The detection of the aforementioned compounds can be realized by fabricating a polymer-coated tapered optical fiber (TOF), where the polymer works as an absorber, which changes the light propagation conditions in the TOF. The TOFs were manufactured based on a standard single-mode fiber for telecommunication purposes and, as an absorbing polymer, hexafluorobutyl acrylate was used, which is sensitive to organosulfur and organophosphorus compounds. The spectral measurements were conducted in a wide optical range—500–1800 nm—covering the visible part of the spectrum as well as near infrared part in order to show the versatility of the proposed solution. Additionally, detailed absorption dynamics measurements were provided for a single wavelength of 1310 nm. The analyses were conducted for two concentrations of evaporating compounds, 10 µL and 100 µL, in a volume of 150 mL. Additionally, a temperature dependency analysis and tests with distilled water were carried out to eliminate the influence of external factors. The results presented in this article confirmed the possibility to provide low-cost sensors for dangerous and harmful chemical compounds using optical fiber technology and polymers as sensitive materials.

## 1. Introduction

Since the first introduction of optical fibers, the field of optical fiber technology has been in constant development, providing better and better quality fibers for both telecommunication and sensing applications. Optical fibers are excellent candidates for sensors due to their small size and weight, the possibility to reach a high sensitivity to selected external conditions and intrinsic immunity to electromagnetic fields [1,2]. The aforementioned parameters enable sensors based on optical fibers to detect and measure such factors as, e.g., surrounding refractive index, temperature, pressure, mechanical strain, stress or vibrations and even chemical reagents [3,4,5]. To enhance the sensing possibilities of optical fibers, it is possible to combine them with additional functional materials, such as liquid crystals [6], alkanes [7,8], polymers [9], metals [10,11] or graphene compounds [12]. The most common methods of applying these functional materials to optical fibers is by forming an external coating after polishing the fiber [13] or fabricating a taper [14,15,16], or by filling the internal air holes in a photonic crystal fiber [17]. It is also possible to fabricate new microstructures at the end of the fiber—e.g., polymer microtips or microlenses [18,19]—and between two fibers of the same type or of two different types [20,21].

The tapering process is a very well known method of manufacturing optical fiber sensors sensitive to changes in the external refractive index [18]. This process changes the propagation conditions of fibers without the need to draw new fibers with selected properties. The theoretical and experimental investigations of changing the core and cladding diameters, as well as the distribution of the refractive index profile along the taper region [14], showed that there is a possibility to produce devices combined with an external material—e.g., polymers.

Polymer materials have been used as sorbents for a long time and they are also used for the preparation of coatings. Commercially available polymer fiber coatings are currently among others: polydimethylsiloxane (PDMS), polyethylene glycol (PEG, Carbowax), molecularly imprinted polymers and various polyacrylates (PA) [22,23,24,25].

One of the main challenges in the field of sensing applications is to construct small and efficient sensors capable of working in toxic environments without interaction with the measured factor. Combining an HFBA polymer with an optical fiber is one of the possibilities to create this kind of sensor, which could be able to detect different toxic compounds—especially chemical warfare agents (CWAs). The CWAs are toxic chemical compounds with chemical and physical properties created for military use. Their characteristic feature is the lethal or harmful effects they have on living organisms (humans, animals and plants). The purpose of their use is also to contaminate the atmosphere, land, industrial facilities, vehicles and military infrastructure, crops, etc. The use of such agents was banned in 1997, when the Chemical Weapons Convention was introduced [26].

One of the best known and most frequently used chemical warfare agents is bis (2-chloroethyl) sulfide, also known as sulfur mustard. It belongs to the group of blistering agents; although it usually does not kill, it causes extensive burns. Due to the potential health hazards and legal issues, imitations such as 2-chloroethyl ethyl sulfide (2-CEES) or thioxane (THX) are commonly used. They have similar structures and physicochemical properties but are less toxic and less harmful to environment than sulfur mustard. Another known and once-used agent is sarin, an organophosphorus compound belonging to the paralytic-convulsive agents that act on the human nervous system as powerful neurotransmitters or vice versa, quickly blocking the action of natural neurotransmitters. These agents are the most dangerous and, at the same time, the most effective poisonous warfare agents. Among the imitations of sarin currently used, trimethyl phosphate (TMP) and dimethyl methylphosphonate (DMMP) are the most frequently used [27]. Among the currently used methods of detecting chemical warfare agents are those, referred to as military, intended for use directly on the battlefield/at the site of contamination.

Such devices include: indicator tubes, spectrophotometric contamination indicators or ion mobility spectrometers. In laboratory conditions, gas or liquid chromatography with mass spectrometry, Fourier-transform infrared spectroscopy and nuclear magnetic resonance are usually used. However, these are methods that require appropriate sample preparation, e.g., by solid phase microextraction of gaseous products [27,28,29,30].

The promising sorption properties of some of the acrylates presented during the analysis of the aforementioned CWA simulants [31,32] inspired this research. The agents selected for this research were TMP, which is a simulant of organophosphorus compounds, such as sarin, and THX, which is an organosulfur compound that is an imitation of sulfur mustard.

In our paper, tapered optical fibers (TOFs) covered in a polymer—hexafluorobutyl acrylate (HFBA)—layer were prepared to be utilized as a base element for the detection and measurement of the aforementioned organosulfur and organophosphorus compounds imitating CWAs. The optical fiber tapers were produced by means of a low-pressure flame technique [14]. The choice of HFBA was dictated by its absorption and optical properties. Most acrylates can be used to sorb CWAs depending on the admixtures that enable this sorption. Working in fiber optic technology, we must additionally ensure beam propagation—focusing not only on chemical but also optical properties. The parameters presented in Table 1, particularly the refractive index, fulfil the propagation conditions (the refractive index of cladding should be lower than the refractive index of the core), including the minimization of optical losses during beam propagation in the taper. In addition, the acrylate HFBA fulfills the adhesion properties optical fibers. Additionally, we wanted to use a multifunctional monomer that increases its volume during the polymerization process. We investigated both the possibility of detecting and the sensitivity of the proposed novel sensor based on an optical fiber taper covered by an HFBA layer of organosulfur and organophosphorus compounds. The studies were undertaken in a wide spectral range of 500–1800 nm. The HFBA layer was applied to a previously prepared taper in order to provide the functionality of an absorber of the aforementioned compounds [33]. To cover tapers in HFBA, a deep coating method was used. The changes in spectral characteristics in a wide range for each deposition of HFBA and concentration of measured agents are presented and discussed.

## 2. Materials and Methods

### 2.1. Materials

The investigation of possibilities to detect organosulfur and organophosphorus compounds was performed on one of the most common single-mode fibers used in telecommunication, SMF-28 form Corning. It has 8.2 μm core diameter and standard 125 μm cladding diameter, with a 0.36% refractive index difference and cutoff wavelength of 1260 nm.

The study consisted of two configurations of samples. In the first one, standard TOF with no additional coating was used to determine the reference. In the second one, TOF was covered in a polymer-HFBA. Additionally, to confirm the possibility of detection of the aforementioned organosulfur and organophosphorus compounds, the distilled water evaporation was investigated as a neutral external agent in order to provide full information about the behavior of proposed sensor in real-life conditions.

To prepare the photopolymerizable mixture as monomer, an initiator (working with green light) and co-initiator of the following compounds were used, respectively: HFBA-2,2,3,4,4,4-hexafluorobutyl acrylate, eosin Y-2′,4′,5′,7′-tetrabromofluorescein disodium salt and MDEA-methyl di-ethanolamine. Additionally, TEOS-tetraethoxysilane and VTEOS-vinyltriethoxylsilane, were alkoxysilane precursors. All of the materials were purchased from Sigma-Aldrich and used as received without further purification. To support the sorption, OH–TSO-hydroxy-terminated silicone oil from Henan Jinpeng Chemical Co., Ltd., was used. This photopolymerizable mixture was prepared by mixing HFBA (ca 60—70 wt%) with a photoinitiator (eosin Y, 0.5 wt%) and a co-initiator (MDEA, 8.0 wt%) together with a sorption assistant (OH–TSO, ca 40–30 wt%), TEOS (ca 8–12 wt%) and VTEOS (ca 3–6 wt%). All substrates were mixed together in an opaque vial using a vibrating shaker for about 1 min and then the vial was placed in a refrigerator to prevent polymerization by exposure to daylight. In Table 1, the main parameters of HFBA are summarized [33,35].

As the liquid material cannot be evenly distributed around the TOF to perform its sensing function, it must be polymerized. The hardening of liquid mixture by means of photopolymerization creates a stable, thin layer that absorbs external factors and interacts with the beam inside the structure. The photopolymerization process of a mixture with HFBA used in our research is presented in Figure 1.

The basic physicochemical properties of the two compounds used in our study—TMP and THX—are presented in Table 2 and Table 3, respectively [36,37,38]. TMP is a simulant of organophosphorus compounds, such as sarin, and THX of organosulfuric compounds, such as sulfur mustard. Both compounds are highly volatile compounds. The equilibrium liquid gas is achieved after about 60 min and TMP has a faint, slightly irritating odor, whereas THX has a strong, irritating odor.

### 2.2. Preparation and Measurment Methods

Optical fibers consist of two concentrically arranged dielectrics: the core and the cladding. The core is characterized by a higher reflective index than the cladding [39]. This determines the light propagation in that structure based on a total internal reflection. The light is propagated in the form of modes in core and a small part of energy penetrates the cladding, named the evanescent field. In fact, there is no possibility to interact with light inside the core. In order to affect the propagating electromagnetic field, a partial removal of the cladding may be applied in the form of mechanical polishing or chemical etching [39,40,41]. In our research, we utilized another approach-optical fiber taper technology. The manufacturing of a TOF was based on a heating process in a low-pressure flame technique at a temperature of about 1100–1200 °C (softening temperature of glass) and stretching the fiber simultaneously in both directions at the same time to reduce diameter of the fiber and ensure an adiabatic shape of the taper with low internal losses. With a decreasing diameter of the core and cladding, a simultaneous increase in the mode field of light propagating in the structure is achieved [7]. As a result, the field of the fundamental mode occupies an increasingly larger cross-section of the cladding until it reaches the point where it is guided in its entire cross-section. The scheme of a TOF with marked mode propagation is presented in Figure 2.

After the elongation of the optical fiber, the whole structure can be divided into two regions—taper waist and transition part connected with the untapered fiber. In the taper waist region, the propagating field fills the whole cross-section of the structure and also part of it is propagating outside. To minimize the losses, an adiabatic type of taper was prepared—characterized by low insertion loss and a long and even transition region.

The change in propagation conditions is also directly related to the change in optical fiber dimensions. The light no longer propagates at the core/cladding interface, but at the cladding/air interface. In this case, in the TOF the propagating electromagnetic field can be influenced by external conditions, as it is propagating in the whole structure (as a core) and the surrounding medium becomes the cladding. Therefore, in the taper waist region, the propagating field is sensitive to changes in the external reflective index of the material that surrounds the structure. This kind of structure with various materials is widely used to build sensors, filters or amplifiers [42].

The TOFs were manufactured using the fiber optic tapered element technology (FOTET) system, which was widely described in the literature [6,7,8]. The FOTET system uses a movable torch based on a gas mixture (propane–butane-oxygen) to obtain the required optical fiber elongation temperature as well as to heat the chosen length of fiber during the process of elongation. The flame is obtained in a low-pressure technique. The temperature of the flame depends on a gas mixing ratio which is set by gas regulators and oscillates within the limits of glass softening. The obtained tapered optical fibers were characterized by an elongation of 25.05 ± 0.05 mm, losses of 0.14 ± 0.06 @ 1550 nm and a taper waist diameter of 7.5 ± 0.5 μm. These parameters were chosen from preliminary studies and the experience of the research team.

After manufacturing, the TOFs were properly secured in a glass V-grove with a UV-curable glue (NOA 81) at both ends of the untapered region, as shown in Figure 3, to avoid mechanical destruction and make it available for the proper application of the polymer mixture. At the same time, the protective plate allowed interaction with TMP and TXH compounds during measurements.

The process of covering the TOF with the HFBA mixture was provided using the deep coating method. In order to deposit the HFBA coating on a taper, first it was thoroughly cleaned with isopropanol to remove any dust particles. Then, the liquid polymer mixture was deposited directly on a taper waist with a syringe, forming a layer around the taper. In the next step, the HFBA layer was polymerized using external green light (532 nm). Our experiments showed that conducting the polymerization through the optical fiber did not fully cure the polymer; therefore, the polymerization took place from the outside with a laser beam, which covered the area of and optical fiber taper waist with polymer. For the deep coating process, we used a laser with a wavelength of 532 nm and an average power of 25 mW, which allowed us to achieve full photopolymerization after 3 min.

The unpolymerized excess of the mixture was then removed with isopropanol.

The scheme of a TOF and its cross-section with additional HFBA coating as well as evaporating reagents is presented in Figure 4.

The spectral measurements were performed in ranges of 550–1200 nm and 1200–1800 nm. As a light source, we used a supercontinuum source SuperK Extreme from NKT Photonics which covered both the aforementioned ranges. The supercontinuum was introduced into the one end of the optical fiber with a taper and the second end was connected to the detector. This configuration allowed us to perform measurements of the spectral distribution of power transmitted through the system. As a detector, we used two optical spectrum analyzers form Yokogawa-AQ6375 for the range 1200–2400 nm and AQ3673 for the range 350–1200 nm. The scheme of the spectral measurements system and its picture is shown in Figure 5.

Due to the limited ability to record the above-mentioned method in time and the need to examine changes from the moment of placing the tested material under the taper, a second configuration was built that allows for power measurements in time for a single wavelength. As a source, the fiber-coupled DFB laser S3FC 1310 nm from Thorlabs was used. The length of the emitted light corresponds to the lowest losses and single-mode propagation for the SMF28 fiber. The light was introduced by FC/PC connector spliced to the one end of the taper. As a detector, a photodiode power sensor S144C connected to the digital Optical Power and Energy meter PM100D from Thorlabs was connected by the second end. To observe the deposition of polymers on TOFs, the scanning electron microscope (SEM) Phenom G2 Pro from FEI company was used. For each of the cases: for distilled water, TMP and THX, in the amounts of 10 µL and 100 µL, were placed on a glass plate under the taper and covered with a pan, ensuring the achievement of the desired concentration of 150 mL from the evaporation.

## 3. Results

For both configurations of samples, standard TOF with no coating and TOF covered in HFBA—we prepared separate consecutive series of measurements. At first, we focused on the influence of the studied agents on an uncoated TOF, then established the temperature stability of the TOF covered in HFBA and its response to exposition to pure water (H_2_O) as an inert factor, and finally we performed studies on TMP and THX’s influence on the proposed sensor. With this workflow, it was possible to obtain reference measurements and determine the influence of each modification. The investigations were performed in a wide spectral range to observe the influence of the aforementioned organosulfur and organophosphorus compounds on light propagation through the taper and its impact over time—immediately after placing the reagents under the sample, after half an hour, after one hour, after two hours and after 24 h.

### 3.1. Sensor Preparation

The SEM images of the optical fiber’s cross-section and taper waist region are presented in Figure 6a,b, respectively. Additionally, in Figure 6c, a picture of a final TOF secured on a protective glass is shown. As was mentioned before, the diameter of a taper waist in fabricated TOFs was equal to 7.5 ± 0.5 μm. The size of the whole sensor with the protective glass was approximately 50 mm.

During the manufacturing process, the optical losses were controlled in a setup integrated with FOTET system. Only TOFs with transmission losses below 0.2 dB @ 1550 nm were used in further studies. The SEM images of TOFs covered in HFBA are presented in Figure 7.

An optical fiber taper covered in HFBA can be seen in Figure 7a,b, along with the dimensions of chosen part: the covering taper by HFBA created on all the taper thin layer had dimensions of about 0.5 µm with irregular thickenings (spot of polymerized droplets), whose dimensions are bigger than the former by over 2 times, below 1.5 µm.

### 3.2. Uncoated TOF

In this part, we present the spectral measurements of the influence of evaporation of the two studied compounds (TMP and THX) on an uncoated TOF.

The spectral characteristics of the influence of the TMP and THX agents for different times of evaporation in a range of 500–1800 nm are presented in Figure 8 and Figure 9, respectively.

As can be observed in Figure 8 and Figure 9, the exposure of an uncoated TOF to vapors of TMP and THX agents did not influence the light propagating through the taper. We observed that glass did not react with these compounds, which is confirmed by our research. Slight differences in the transmitted power level were found—increases in all ranges, especially in the case of THX for wavelengths above 1400 nm, may come from supercontinuum source power fluctuation during the measurement—as well as changes in the external environment around taper—a change in the refractive index, which influences longer waves.

The results presented in this subsection may be treated as a reference point for further analyses. We confirmed that uncoated TOF is insensitive to both to TMP and THX agents, and in order to allow the detection of these compounds, it needs to be enhanced by covering it in a specially designed polymer.

### 3.3. HFBA-Coated TOF

In the second part of our measurements, we investigated thermal stability and response to exposition to the inert factor of TOFs coated in HFBA. Thermal stability was studied by measuring the spectral power distribution of a light transmitted through the system placed in a thermal chamber. The measurements were performed in an external temperature range of 0–50 °C, which covers the range of natural conditions. As the inert factor, a pure distillated water was used. The results of thermal analysis of HFBA-coated TOF are presented in Figure 10.

As can be seen in Figure 10, even with additional irregular deposition of HFBA, the taper covered with this polymer remained insensitive to external temperature in the whole analyzed range. This result allows for temperature-independent measurements of real organosulfur and determines organophosphorus compounds’ influence on light propagation, without additional influence of external factors.

In addition to the ever-changing temperature, another one of the most important factors in natural conditions is humidity. Therefore, for the proper characterization of a sensor sensitive to external chemical vapors, we had to determine if it is also sensitive to evaporating water. To perform this task, we used vapors of distillated water in two different temperatures—room temperature (about 21 °C) and near water boiling temperature (about 95 °C). The results of time-dependent power for wavelength of 1310 nm transmitted through an HFBA-coated TOF under the influence of evaporating water are presented in Figure 11.

As can be seen in Figure 11a, the influence of water evaporation at room temperature on the power transmitted through a HFBA-coated TOF is negligible. The slight increase in the power level is at the order of 0.1 mW (>10%) and may be attributed to the properties of the used laser source S3FC1310, for which power stability across warm-up in time is described by the producer as 15 min: ±0.05 dB @ 25 °C. For vapors of water near boiling temperature (Figure 11b), a rapid change in transmitted power was observed when the water was placed under the sample. After about 2 min, the transmitted power stabilized, although it did not return to the initial level. This effect is related to the deposition of evaporating water on a taper and changing the RI around the tapered region waist as well as the transition region. Additionally, for boiling water, condensation can occur in a taper region, slightly changing the propagation conditions.

### 3.4. TMP Detection

In the next part, we focused on the possibility of TMP detection and its influence on a field propagating in the TOF-based sensor. To ensure clarity and consistency in our studies, we present results for two configurations of samples—for uncoated TOF and for TOF covered in HFBA. The spectral results of the power transmitted through each system under the influence of TMP vapors for 1 h and 24 h are presented in Figure 12 and Figure 13, respectively (the volume of TMP used in the study was 100 µL). As a reference, the taper covered with HFBA is presented.

As can be seen in Figure 12 and Figure 13, the evaporating TMP influences the HFBA-coated TOF-based sensors in such a way that the power transmitted through the system increases significantly, especially in the IR range, in a mostly uniform way in the whole studied spectral range compared to the TOF with HFBA before being exposed to the agents. The power increase is directly connected to the fact that TMP vapors are being absorbed by the HFBA deposited along the taper waist, resulting in the change in the polymer’s refractive index, which directly influences the field propagating through the taper. This interaction is evidenced by the fact that the power transmitted through the system increases, while part of the electromagnetic field is being emitted out of the taper. It is more evident in an IR range, as shown in Figure 12b and Figure 13b. This is connected with the properties of the SMF28 fiber, which was projected for single-mode propagation of IR range. For the visible range, a multimode propagation was obtained, which can reduce the observed phenomenon.

### 3.5. THX Detection

In the next part, we focused on the possibility of THX detection and its influence on a field propagating in the TOF-based sensor. Again, we present results for all three aforementioned configurations of samples. The spectral results of the power transmitted through each system under the influence of THX vapors for 1 h and 24 h are presented in Figure 14 and Figure 15, respectively.

As can be seen in Figure 14 and Figure 15, the evaporating THX also influences the HFBA-coated TOF-based sensors in such a way that the power transmitted through the system decreases significantly, especially in an IR range in a mostly uniform way in the whole studied spectral range compared to the TOF with HFBA before being exposed to the agents. The power decrease is directly related to the fact that THX vapors are being absorbed by the HFBA deposited along the taper waist, resulting in the change in the polymer’s refractive index, which directly influences the field propagating through the taper. Compared to TMP, it is easy to observe that THX causes an increase in the RI, simultaneously changing the light propagation condition, while part of the electromagnetic field is emitted out of the taper. It is more evident in an IR range like the ones shown in Figure 14b and Figure 15b. This is connected to the properties of the SMF28 fiber, which was projected for the single-mode propagation of the IR range. After 24 h, for the visible range the answer of the sensor is imperceptible. These effects can be result of fiber propagation properties which works for the visible wave range as a multimode. Due to THX’s properties, there can be also process of desorption of the investigated THX materials from polymer to the state for which propagation parameters are not different enough from pure polymer.

### 3.6. Detection Dynamics

The next step was connected with time-dependent measurements of transmitted power changes for a single wavelength of 1310 nm, which corresponds to the SMF operating wavelength (optimization of the sensor for single-mode propagation with low propagation losses). The results of these measurements for two volumes of TMP and THX are presented in Figure 16.

As can be seen in the presented results, the characteristics of the obtained signals are highly dependent on the amount of the evaporating agent. A smaller amount of material (10 µL) of TMP and THX evaporating in the volume of 150 mL causes a faster evaporation of the whole material, which leads to this process being completed in a shorter amount of time. This process causes the sorption (deposition) of the material on the HFBA surface, creating a layer of a material with the refractive index higher than that of the polymer. According to the wave propagation principle, the light beam propagates both in the taper and leaks into the polymer with the absorbed material (TMP or THX). In the case of the TMP, the rapid initial process may cause the leakage of the propagating field out of the structure of taper-polymer-absorbed agents. Further exposure to the TMP causes the entire volume of HFBA to absorb the agent, therefore changing its refractive index so that the light may be guided again in the taper and the transmitted power slowly increases. A similar situation occurs for THX, where a rapid change in the transmitted power is also visible, although the power level drops by about 50% and fluctuates around this value. In this case, during the initial phase almost all THX vapors are absorbed by the HFBA, changing its refractive index, and the system stays in a long-term equilibrium.

With a 10-fold increase in the amount of absorbed material (to 100 μL), the changes in the power are not as rapid as before due to the slower evaporation rate of the agents. A larger amount of liquid material and thus constant evaporation rate eventually lead to either an improvement in the guiding conditions for TMP or a complete loss of guiding properties for THX. In the first case, the slow sorption of TMP changes the HFBA refractive index in a way that enhances the mode field confinement in the TOF-based structure, which is visible in Figure 16b as the increase in the power transmitted through the system. In the case of the THX agent, the slow absorption of its vapors by HFBA causes a constant increase in the polymer’s refractive index. This process manifests itself as an almost linear decrease in the transmitted power—up to the point where all of the field is leaking outside of the structure and no more light is being transmitted through the system.

During the course of the measurement process, slight spontaneous desorption was possible when the concentration of vapors was too high in relation to the sorption capacity of the absorbent material, from which small power fluctuations might occur. The results previously obtained for distilled water (in Section 3.3) are also presented in Figure 16 to provide a reference level to an inert agent. Additionally, as was presented in both plots in Figure 16, the laser source power level was not constant over time, which might cause some fluctuations of the measured signal for all samples.

As can be seen in Figure 16, the comparison of power dynamics measurements of HFBA-covered TOF with TMP and THX vapors confirmed that it is possible to detect different CWAs using the proposed sensor. The changes in power transmitted through the described structure are significantly different not only for different agents but also for different amounts of evaporating material, which may imply some limitations on the applicability of such sensors. Still, we believe that the presented possibility for the detection of any amount of harmful organosulfur or organophosphorus agents is an important step forward towards universal anti-warfare monitoring systems.

## 4. Conclusions

The results presented in this paper the confirmed the possibility of manufacturing low-cost CWAs sensors based on TOF covered with HFBA. The proposed sensors were tested in a wide optical range, 500–1800 nm, providing the possibility of detecting both TMP and THX. The HFBA polymer absorbs organosulfur and organophosphorus compounds and changes its optical parameters, such as the refractive index and material composition, which influence the light propagation in a TOF. The response of such sensors changes according to a spatial volume of external organosulfur and organophosphorus compounds as well as for another materials, which can evaporate. For all kinds of materials, we obtained different results, allowing us to distinguish between these compounds. The differences are connected to the rate of material evaporation and therefore the dynamics of absorption by HFBA. Based on the presented results, it would be possible to crate a catalog of chemical compounds and their impacts on the light propagating in a proposed sensor. The HFBA-coated TOFs can be used both for CWA detection over a short period (below 1 h) or for long-term observations—even over 24 h. The results psresented in this article confirmed the possibility to provide a sensor for dangerous and harmful chemical compounds, namely organosulfur and organophosphorus compounds, using optical fiber technology and a polymer as an absorbing and sensitive material. The proposed sensor is low cost and easy to manufacture and operate, and therefore may find application not only for military purposes, but also in everyday life, e.g., in terrorist attack monitoring and warning systems.

## Figures and Tables

**Figure 1 polymers-14-00612-f001:**
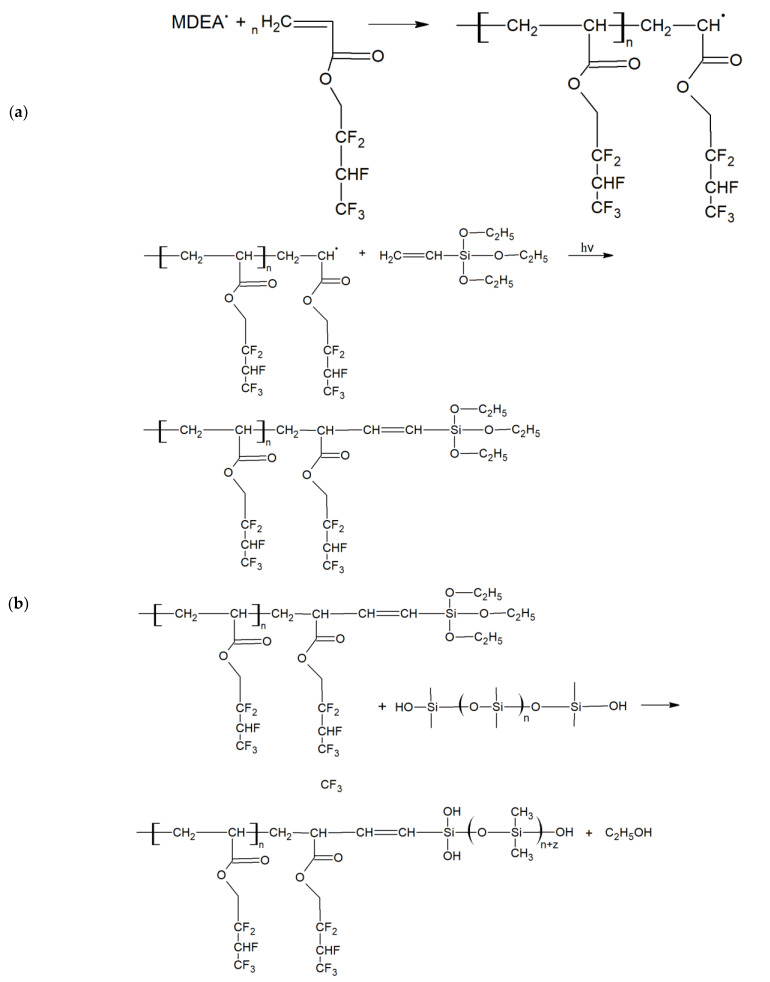
Examples of courses of a photopolymerization reactions: (**a**) with green laser light (532 nm) initiation; (**b**) HFBA mixture polymerization.

**Figure 2 polymers-14-00612-f002:**
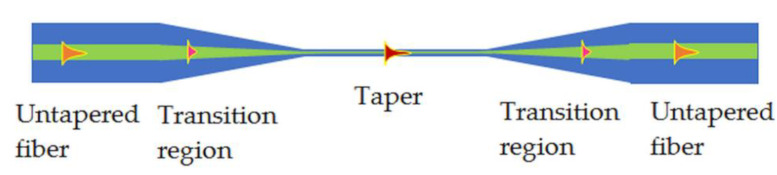
Scheme of a TOF structure and mode propagation.

**Figure 3 polymers-14-00612-f003:**
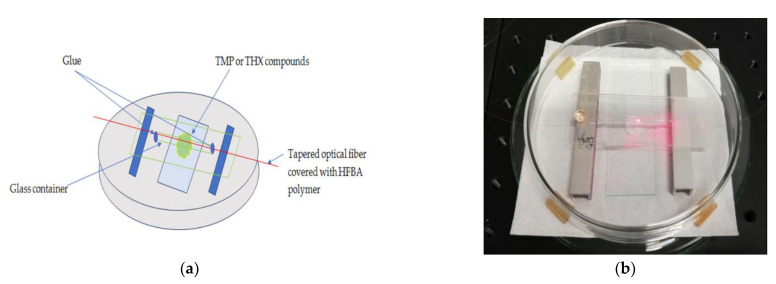
Scheme of a manufactured TOF secured on a glass plate (**a**) and picture of measured TOF (**b**).

**Figure 4 polymers-14-00612-f004:**
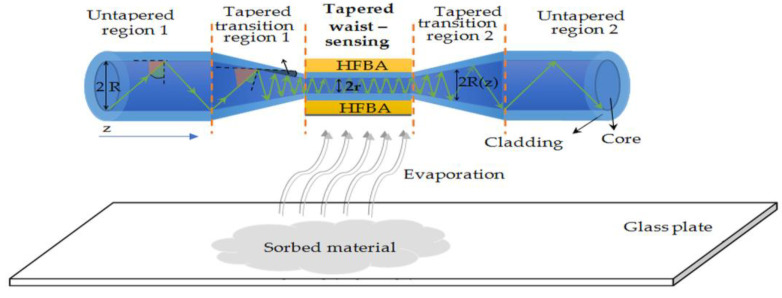
Scheme and cross section of optical fiber taper with marked superimposed layers of different materials.

**Figure 5 polymers-14-00612-f005:**
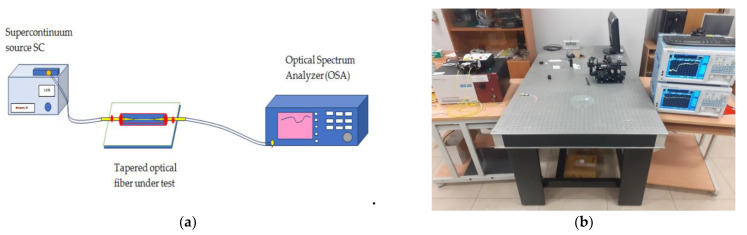
(**a**) Scheme of measurement system and (**b**) picture of the setup.

**Figure 6 polymers-14-00612-f006:**
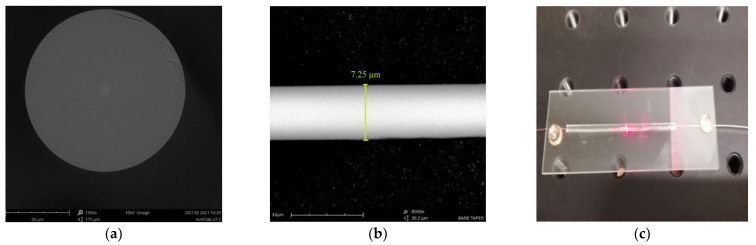
(**a**) Cross-section of SMF fiber; (**b**) taper waist region; (**c**) picture of a final TOF.

**Figure 7 polymers-14-00612-f007:**
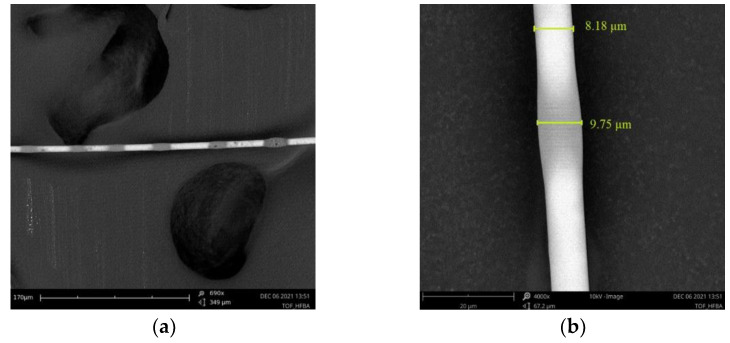
The SEM images of the TOF: (**a**) covered in HFBA (**b**); magnification of covered TOF with HFBA.

**Figure 8 polymers-14-00612-f008:**
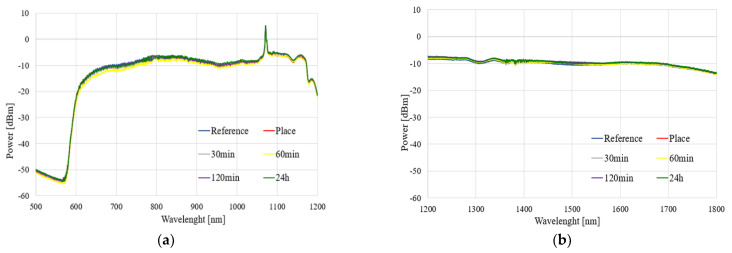
Spectral characteristics of the influence of the TMP agent on transmission through uncoated TOF in the ranges of (**a**) 500–1200 nm and (**b**) 1200–1800 nm.

**Figure 9 polymers-14-00612-f009:**
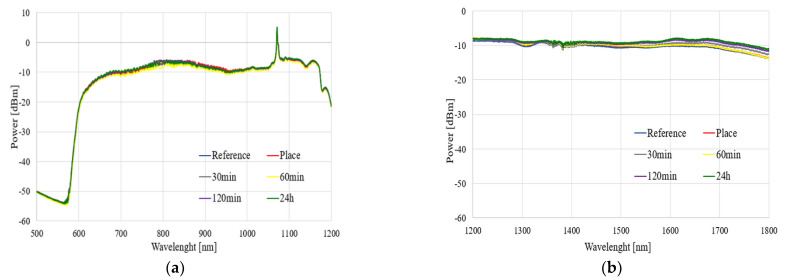
Spectral characteristics of the influence of the THX agent on transmission through uncoated TOF in the ranges of (**a**) 500–1200 nm and (**b**) 1200–1800 nm.

**Figure 10 polymers-14-00612-f010:**
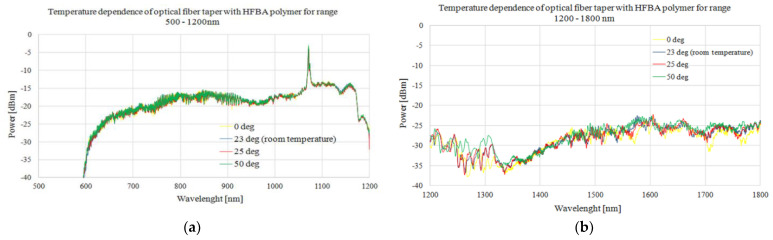
Spectral characteristics of a transmission through HFBA-coated TOF for different temperatures in ranges of (**a**) 500–1200 nm and (**b**) 1200–1800 nm.

**Figure 11 polymers-14-00612-f011:**
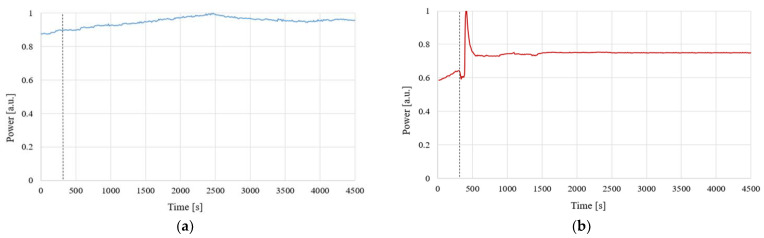
Power response on evaporation of distilled H_2_O—room temperature (**a**) and in boiling temperature (**b**)—on light propagation on TOF covered with HFBA.

**Figure 12 polymers-14-00612-f012:**
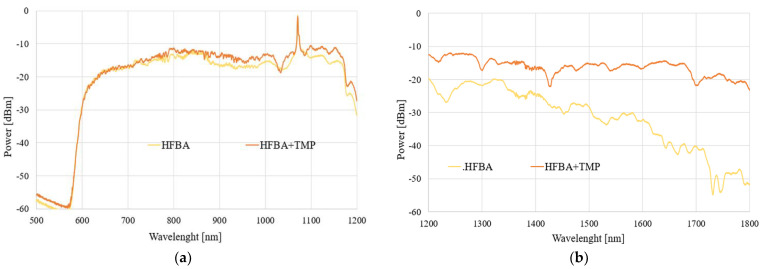
Spectral characteristics of TOF-based systems exposed to TMP vapors for 1 h in wavelength ranges of (**a**) 500–1200 nm and (**b**) 1200–1800 nm.

**Figure 13 polymers-14-00612-f013:**
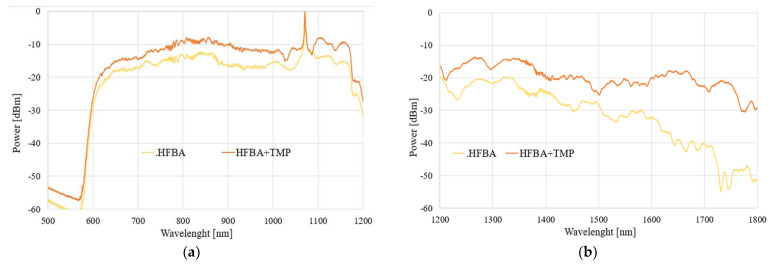
Spectral characteristics of TOF-based systems exposed to TMP vapors for 24 h in wavelength ranges of (**a**) 500–1200 nm and (**b**) 1200–1800 nm.

**Figure 14 polymers-14-00612-f014:**
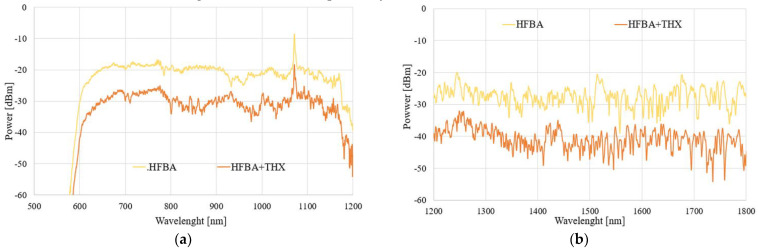
Spectral characteristics of TOF-based systems exposed to THX vapors for 1 h in wavelength ranges of (**a**) 500–1200 nm and (**b**) 1200–1800 nm.

**Figure 15 polymers-14-00612-f015:**
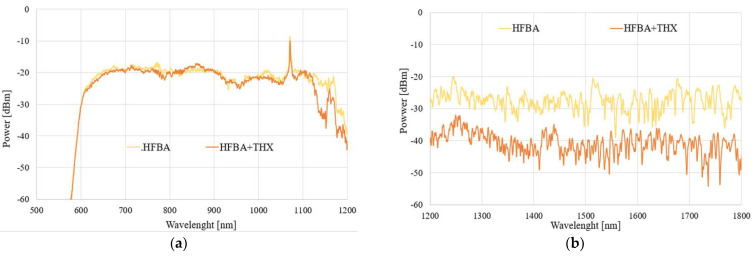
Spectral characteristics of TOF-based systems exposed to THX vapors for 24 h in wavelength ranges of (**a**) 500–1200 nm and (**b**) 1200–1800 nm.

**Figure 16 polymers-14-00612-f016:**
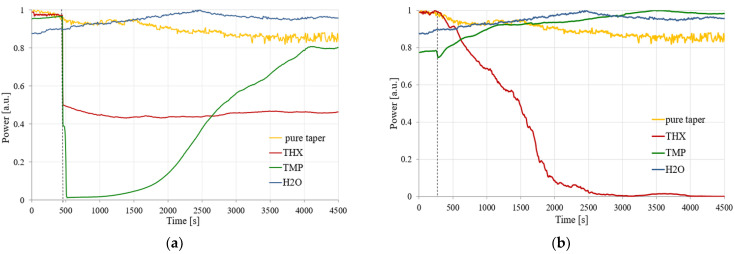
Comparison of the HFBA polymer’s interaction with TMP and THX agents: time measurement for 1310 nm wavelength—10 µL (**a**) and 100 µL; (**b**) in a 150 mL spatial volume.

**Table 1 polymers-14-00612-t001:** Single-mode fiber and HFBA properties [34,35].

Single-Mode Fiber Properties		
Attenuation	≤0.32 @ 1310 nm	dB/km
Core diameter	8.2	μm
Mode Field Diameter	9.2 ± 0.4 @1310 nm	µm
Cladding diameter	125 ± 0.7	μm
Effective Group Index of Refraction	1.4677 @ 1310 nm	---
Refractive Index Difference	0.36	%
HFBA Properties		
Monomer refractive index	1.352	-
Mixture refractive index	1.379	-
Density of monomer	1.389	g/mL
Boiling temperature	40–43 @8 mmHg	°C

**Table 2 polymers-14-00612-t002:** TMP properties [36,37].

Parameter	Value	Units
Boiling point	197.2	°C
Density	1.2144	g/mL
Refractive index	1.3967	

**Table 3 polymers-14-00612-t003:** THX properties [36,38].

Parameter	Value	Units
Boiling point	147	°C
Density	1.114	g/mL
Refractive index	1.5095	

## Data Availability

The data presented in this study are available on request from the corresponding author.

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
