# Peer review of "Detection of Organosulfur and Organophosphorus Compounds Using a Hexafluorobutyl Acrylate-Coated Tapered Optical Fibers"

_polymers, 2022, doi:10.3390/polym14030612_

Round 1
Reviewer 1 Report
In this manuscript, the authors presented a study to test the possibility of detection of organosulfur and organophosphorus compounds with polymer-assisted optical fiber. A polymer-coated tapered optical fiber (TOF) was coated with hexafluorobutyl acrylate (HFBA) to serve as an absorber for the organic materials. After absorbing the organic material, the light propagation conditions in the TOF, and was sensitive to organosulfur and organophosphorus compounds. The spectral measurements were conducted in a wide optical range from 500-1800 nm to show versatility of proposed detection mechanism. The sensors showed good responses to the testing fluids with obvious light absorbance changes in the infrared range. Overall, the manuscript was organized well and their findings did bring some insights for organic compound detection. But some details should be provided before publication. Some suggestions are listed below for the authors to perfect the manuscript.
- In the paragraph from line 83-86, it is not quite clear why the authors choose HFBA. Please provide more detail on the functionality of HFBA to the CWAs. It would be eaier for the readers to understand the reason of using this polymer as a detective coating. Moreover, the citation in line 95 has an incorrect number.
- Please list the optical properties of the optical fiber, and compare with the HFBA in table 1.
- The HFBA coating on optical fiber was uneven. Did the size of the HFBA beads the same for all the sensing area?
- The compositions or concentrations of the testing fluids in figure 16 (a) and (b) should be written in the figure caption. It is quite confusing for readers to understand: the responses were different for the same compound.
- The detection limits for TMP or THX using this method should be given.
- Why choosing the signal of 1310 nm in section 3.6?
- How was the selectivity of the prepared sensors to TMP or THX? What happens if one mix TMP and THX together?
Reviewer 2 Report
Good work! I would like to ask why using TMP and not DMMP which is a more apt simulant for Sarin and other G-based agents. I think DMMP could also provide better results as its bp is lower than that of TMP. In any case, no need to repeat these experiments.
On the other hand, the simulant for HD (sulfur mustard) used is THX and that is a very different compound altogether with very different reactivity, so I am not sure how relevant those findings are with your system. This obviously opens the question on selectivity for the method, can any chemical with a decent vapor pressure give curves similar to those described in the paper?
The work is good and introduces this approach as a platform for further evaluation.
Round 2
Reviewer 1 Report
The authors have revised the manuscript according to review comments with acceptable revision.